# Characterization of RARRES1 Expression on Circulating Tumor Cells as Unfavorable Prognostic Marker in Resected Pancreatic Ductal Adenocarcinoma Patients

**DOI:** 10.3390/cancers14184405

**Published:** 2022-09-10

**Authors:** Christine Nitschke, Benedikt Markmann, Marie Tölle, Jolanthe Kropidlowski, Yassine Belloum, Mara R. Goetz, Hartmut Schlüter, Marcel Kwiatkowski, Marianne Sinn, Jakob Izbicki, Klaus Pantel, Cenap Güngör, Faik G. Uzunoglu, Harriet Wikman

**Affiliations:** 1Department of General, Visceral and Thoracic Surgery, University Hospital Hamburg-Eppendorf, 20246 Hamburg, Germany; 2Mildred Scheel Cancer Career Center, University Hospital Hamburg-Eppendorf, 20246 Hamburg, Germany; 3Institute of Tumor Biology, University Hospital Hamburg-Eppendorf, 20246 Hamburg, Germany; 4Institute of Clinical Chemistry and Laboratory Medicine, University Hospital Hamburg-Eppendorf, 20246 Hamburg, Germany; 5Institute of Biochemistry and Center for Molecular Biosciences (CMBI), Leopold-Franzens University Innsbruck, 6020 Innsbruck, Austria; 6II. Medical Clinic and Polyclinic (Oncology), University Hospital Hamburg-Eppendorf, 20246 Hamburg, Germany

**Keywords:** liquid biopsy, circulating tumor cells, RARRES1, pancreatic ductal adenocarcinoma, biomarker

## Abstract

**Simple Summary:**

Our explorative study used a microfluidic-based approach for circulating tumor cell (CTC) detection in 55 pancreatic ductal adenocarcinoma (PDAC) patients before treatment initiation (baseline) and during follow-up (FUP). For the first time, we assessed the expression of retinoic acid receptor responder 1 (RARRES1) on CTCs. CTCs were detected in 25.5% of patients at baseline, while the detection rate during FUP was higher (45.5%). Especially high CTC counts during FUP in resected patients were associated with early tumor relapse (*p* = 0.02). Combining CTC detection and RARRES1 protein expression showed that RARRES1-positive patients with high CTCs counts after curative operation during FUP had a worse prognosis (*p* = 0.001). In conclusion, RARRES1 is a new marker of interest for further research investigations on subtypes of CTCs in PDAC.

**Abstract:**

Background: In pancreatic ductal adenocarcinoma (PDAC), the characterization of circulating tumor cells (CTCs) opens new insights into cancer metastasis as the leading cause of cancer-related death. Here, we focused on the expression of retinoic acid receptor responder 1 (RARRES1) on CTCs as a novel marker for treatment failure and early relapse. Methods: The stable isotope labeling of amino acids in cell culture (SILAC)—approach was applied for identifying and quantifying new biomarker proteins in PDAC cell lines HPDE and its chemoresistant counterpart, L3.6pl-Res. Fifty-five baseline and 36 follow-up (FUP) peripheral blood samples were processed via a marker-independent microfluidic-based CTC detection approach using RARRES1 as an additional marker. Results: SILAC-based proteomics identified RARRES1 as an abundantly expressed protein in more aggressive chemoresistant PDAC cells. At baseline, CTCs were detected in 25.5% of all PDAC patients, while FUP analysis (median: 11 months FUP) showed CTC detection in 45.5% of the resected patients. CTC positivity (≥3 CTC) at FUP was significantly associated with short recurrence-free survival (*p* = 0.002). Furthermore, detection of RARRES1 positive CTCs was indicative of an even earlier relapse after surgery (*p* = 0.001). Conclusions: CTC detection in resected PDAC patients during FUP is associated with a worse prognosis, and RARRES1 expression might identify an aggressive subtype of CTCs that deserves further investigation.

## 1. Introduction

Pancreatic ductal adenocarcinoma (PDAC) is still one of the deadliest cancer entities since it is often diagnosed late due to unspecific symptoms, its aggressive nature, and its early dissemination. In addition, current clinically available tools are inadequate for an early diagnosis, and therapeutic options are limited [1]. As most patients have already progressed to a late stage at diagnosis, only the minority of patients qualify for a curative intended surgery with consecutive adjuvant chemotherapeutic treatment [2,3,4,5].

In recent years, liquid biopsy has gained importance as a potential alternative source of tumor material for molecular diagnostics. Analyses in blood—such as circulating tumor cells (CTCs)—represent most malignant cells [6]. Therefore, tremendous clinical interest has emerged due to the minimally invasive nature of liquid biopsy, allowing sequential sampling. It has become an important alternative tool for cancer diagnostics and prognosis of poor survival [6,7].

The potential importance of early detection and characterization of CTCs in PDAC patients was shown in a PDAC mouse model, in which CTCs with metastatic potential were already shed during the formation of the primary tumor before it became detectable by histological methods [8]. CTC detection in peripheral blood of patients with PDAC has been assessed as a diagnostic option in several studies, with most studies focusing on metastatic or locally advanced tumors [9,10]. Even in the cohorts that included patients with all tumor stages, metastatic cancer stages were the most represented ones (>80%) [11]. In these previous studies with study cohorts from 14 to 172 PDAC patients, CTC positivity rates varied significantly between different enrichment methodologies (7–42% in curative patients and 19–48% for palliative patients) [6,12,13,14]. CTCs have been shown to be independent predictors of poor prognosis in most of these studies. They could be helpful for patient stratification in addition to classical parameters, such as imaging, TNM classification, and Ca19-9 levels [12,13,14]. Furthermore, the impact of CTC detection on poor survival has been supported by two meta-analyses, including more than 600 patients [15,16].

Many previous studies on CTC detection in PDAC patients have used cell surface antigens (positive selection using mainly EpCAM) for CTC isolation. As these methods’ positivity rates have been relatively low, antigen-independent CTC detection methods, such as different microfluidics-based systems, have become of interest [14]. At this point, there is no gold-standard method for CTC detection in PDAC, and comparisons between the existing studies are complicated due to small cohorts, low detection rates, and the focus on metastatic and locally advanced stages [6,9,10]. PDAC CTCs clearly show significant heterogeneity (e.g., co-expressions of keratins and mesenchymal markers, such as vimentin). They are thus not consistently detectable by using only one antigen, such as EpCAM [17].

Despite the CTC detection showing a prognostic impact on poor survival, it does not affect therapeutic strategies [18]. Thus, CTC detection has no use in a daily clinical routine yet [14,18]. Therefore, CTC characterization has become of interest alongside the initial research focus on CTC enumeration [14]. Here, the analysis of additional marker proteins expressed on CTCs in PDAC could be valuable [3,4,5,12,18].

Numerous different cellular and molecular detection methods have been developed with varying success to improve the molecular characterization of CTCs and to detect the most malignant clones with the highest metastatic potential and clonal heterogeneity [14,18]. For this reason, we initially screened for novel PDAC markers for aggressive disease course using a proteomic approach combined with mass spectrometry. As one of the most abundantly expressed markers in the more aggressive chemoresistant cells, we successfully identified the retinoic acid receptor responder 1 (RARRES1) protein. Due to the outstanding expression of the RARRES1 protein in the proteomic approach and mass spectrometry, it was chosen for the combined CTC analysis.

In conclusion, our study analyzed the possible benefit of combining RARRES1 expression—as a relatively unknown but potentially unfavorable prognostic marker—with CTC detection as a companion prognostic marker in PDAC, revealing the potential of proteins with an impact on poor prognosis in aggressive PDAC—as future biomarkers in liquid biopsy analysis.

## 2. Materials & Methods

### 2.1. Cell Lines

The human pancreatic duct epithelial (HPDE) cell line was cultured with keratinocyte-SFM media containing EGF and pituitary extract (Invitrogen, Darmstadt, Germany). L3.6pl-Res and WT cells, BxPC-3, PANC-1, and PANC-2 cells were cultured in RPMI 1640 (Invitrogen) or DMEM supplemented with 10% FCS and 200 IU/mL of Pen-Strep at 37 °C and 5% CO_2_. The development of chemoresistant L3.6pl cells (L3.6pl-Res) was previously published [19]. Briefly, sensitive cells were exposed to an initial gemcitabine concentration of 0.05 µM/L, and the surviving population was grown to 80% confluence and passaged three times over 14 days to ensure stable viability. Cells were treated with sequentially increased gemcitabine concentrations up to the clinically relevant concentration of 2 µM/L for 30 days. L3.6pl-Res cells are ~20 times more resistant to chemotherapy than L3.6pl-WT cells. All used cell lines were genotyped by DNA fingerprinting (Identifiler^TM^-Kit, Thermo Fisher, Bremen, Germany).

### 2.2. Stable Isotope Labeling with Amino Acids in Cell Culture (SILAC)

L3.6pl-Res and L3.6pl-WT cells were cultured in SILAC RPMI 1640 media (Thermo Fisher, Germany) supplemented with 10% dialyzed FCS (Thermo Fisher, Germany) and 200 IU/mL of Pen-Strep (Invitrogen, Germany) at 37 °C and 5% CO_2_. Resistant cells (continuously treated with 2 µM gemcitabine) were metabolically labeled with “heavy” isotope amino acids (^13^C_6_-Arg + ^13^C_6_-Lys). In contrast, the parental-sensitive L.3pl-WT cells were cultured with “light” amino acids according to the manufacturer’s protocol (Thermo Fisher, Germany) for a minimum of 8 split cycles. Following trypsin digestion and mixing of both cellular lysates, the successful incorporation of isotopes into cellular proteomes was quality tested using Liquid Chromatography-Tandem Mass Spectrometry (LC-MS/MS) analyses. Data were further analyzed using Max-Quant software. Protein identification was based on MS/MS spectra, while ratios of the respective SILAC pairs were used for relative protein quantification. Among all the identified proteins, we excluded those with thresholds lower than 5-fold change and a *p*-value of ≤0.05.

### 2.3. RNA Isolation and Real-Time Reverse Transcription-PCR (Real-Time RT-PCR)

Total RNA was isolated using TRIzol^®^ (Invitrogen). The dried pellet was cleaned with the RNeasy MiniElute Kit (Qiagen, Hilden, Germany). RNA-concentration was measured on a NanoDrop^®^ Spectrophotometer (Peqlab, Erlangen, Germany). Real-time RT-PCR was conducted to quantify gene expression (primer sequence: hRARRES1; Forward: 5′-AGGTGTCACACTACTACTTGG-3′; Reverse: 5′AGCTGTTGACAGTGGTACTTC-3′). One microgram of total RNA was reverse-transcribed using the Transcriptor-cDNA-Kit (Roche, Grenzach-Wyhlen, Germany). PCRs were carried out in a Mastercycler^®^ ep-realplex (Eppendorf, Germany). Data were analyzed according to the comparative C_T_ method and normalized for Cyclophilin expression in each sample. PCR primers were ordered from Thermo Fisher, Germany (Hs00161204_m1).

### 2.4. Western Blot

Cells were lysed using RIPA buffer (Sigma Aldrich, Germany) containing a 1× protease inhibitor cocktail (Roche, Germany). Supernatants were quantified and resolved by SDS-PAGE followed by immunoblotting. Primary antibodies (Anti-RARRES1, 1:1000, Abcam, Boston, MA, USA) were incubated overnight (+4 °C), followed by a 2 h incubation with HRP-labeled secondary antibodies (1:5000; Thermo Fisher, Germany).

### 2.5. Immunocytochemistry

L3.6pl-Res cells were used for immunocytochemical analysis. Briefly, cells were grown on sterile coverslips (50–60% confluency) accompanied by their fixation in 4% formaldehyde in PBS for 10 min. After washing, cells were permeabilized using Triton-x (0.1%) and incubated with primary anti-RARRES1 antibody (1:1000, Abcam, USA) overnight, followed by a 2 h incubation with secondary antibodies (1:1000; Alexa^488^, Invitrogen) before mounting. For negative control stainings, only secondary antibodies (Alexa^488^, Invitrogen, Waltham, MA, USA) were incubated overnight. DAPI was used for co-staining nuclei. Image acquisition was performed using confocal microscopy (Leica, Wetzlar, Germany).

#### 2.5.1. Patient Cohort

Our study cohort included 55 patients, of which 36 were curative and 19 palliative adult PDAC patients, being treated at the University Hospital Hamburg-Eppendorf between October 2019 and November 2020. Appendix A shows the number of patients considered for inclusion and matching the inclusion criteria. Clinicopathological data were collected from all patients. The database includes clinicopathological characteristics (e.g., age, gender, TNM classification), operative details (e.g., duration of surgery, type of resection performed, blood loss), and follow-up (FUP) data. The Ethics Committee of Hamburg approved the database and biobank (PV3548).

#### 2.5.2. Blood Collection

EDTA blood samples at volumes of 7.5 mL were taken at baseline before curative surgery (*n* = 36) or palliative (*n* = 19) treatment start. Additional peripheral blood samples (*n* = 47) were drawn three times monthly at FUP visits during and after chemotherapeutic treatment from 22 curative patients.

### 2.6. CTC Detection

We used the marker-independent microfluidic-based Parsortix™ cell separation system in our study to possibly overcome limitations (such as low detection rates) associated with EpCAM-based approaches. Our previous studies have shown that this approach provides size and deformability-based enrichment, which is very suitable for CTC detection in challenging patients [20,21,22].

EDTA blood samples at volumes of 7.5 mL were processed within two hours upon collection by the Parsortix^TM^ system. They were combined with immunofluorescence staining for DAPI, pan-keratins for positive selection, and CD45 for negative enrichment. For immunofluorescence staining, the enriched cell fraction was fixed with 4% paraformaldehyde (PFA, Sigma Aldrich, Burlington, MA, USA) for 10 min at room temperature before being permeabilized with 0.2% Triton X-100 (Sigma Aldrich, USA) for 10 min, blocked with 10% AB-Serum (Bio-Rad, Hercules, CA, USA) and incubated with primary RARRES1 antibody (1:1000; Abcam, Cambridge, UK) at 4 °C overnight. The following day, the secondary antibody Goat Anti-Rabbit (1:1000, Alexa Fluor 488) was incubated for 45 min at room temperature before incubation with DAPI, conjugated cytokeratin [C11 (1:80, Cell Signaling, Danvers, MA, USA), and AE1/3 (1:80, Anti-Pan-Cytokeratin AlexaFluor546 Clone)] (Invitrogen, USA) and allophycocyanin (APC) conjugated CD45 antibodies (1∶150, A647 anti-human CD45 Clone H130 BioLegend) for 45 min. Analysis was performed using immunofluorescence microscopy.

### 2.7. Statistical Analysis

We used SPSS version 26 (SPSS, Inc., Chicago, IL, USA) for statistical analyses. The χ2 test was used to evaluate a potential association between the CTC status and clinicopathologic parameters. Survival curves for patient overall survival (OS) and recurrence-free survival (RFS) were plotted using the Kaplan-Meier method and analyzed by the log-rank test. The OS was the period from the date of curative resection or palliative treatment start to either the date of death or last FUP. The RFS was defined as the period from the date of surgery in resected curative patients to the date of recurrence, last FUP, or date of death, whichever occurred first. The number of at least three CTCs was considered as a high CTC detection.

## 3. Results

### 3.1. SILAC-Based Identification of Abundant RARRES1 in Chemoresistant PDAC Cells

Although new chemotherapy regimens were recently tested and successfully applied to PDAC patients (e.g., FOLFIRINOX), the frequent development of chemotherapeutic resistance severely limits the efficacy of these cytotoxic drugs and is often attributed to aggressive cancer cell progression and ultimately reduced survival. For this reason, we applied a biomarker discovery approach using SILAC, which combines the identification and quantification of new biomarker proteins in chemoresistant and sensitive PDAC cells. In principle, SILAC is a metabolic labeling technique allowing for the isotope (heavy/light amino acids) labeling of total proteomes in living cells. The successfully labeled peptides were subjected to chromatographic separation and high-resolution mass spectrometry (MS) analyses (Figure 1A).

Here, we compared the proteomic profiles of L3.6pl-Res (chemoresistant) and its parental subclone, the L3.6pl-WT cells (chemo-sensitive) [19]. Following trypsin digestion, the MS analyses detected several abundantly expressed peptides in L3.6pl-Res cells (Figure 1A). According to the expression level differences, we excluded those peptides with lower than a 5-fold change (Appendix A). Among the top hits, we identified several peptides with homology to RARRES1 protein (Figure 1B). We further investigated RARRES1 mRNA- and protein expression levels in proof-of-concept studies using western blot and real-time RT-PCR experiments (Figure 1C). Surprisingly, RARRES1 expression was solely detected in chemoresistant L3.6pl-Res cells and was undetectable in normal pancreas cells (HPDE), PANC-1, PANC-2, and BxPC3 cells, suggesting that our SILAC approach was successful. According to the deduced amino acid sequence, RARRES1 is a type I transmembrane protein. To investigate the subcellular localization of RARRES1 in PDAC cells, we also performed immunocytochemistry in L3.6pl-Res cells. RARRES1 staining showed a strong expression in membrane and cytoplasm of L3.6pl-Res cells (Figure 1D).

In agreement the results presented here, an investigation of publicly available protein data from “The Human Protein Atlas” indicated a slightly more unfavorable prognosis for patients with abundant RARRES1 expression in PDAC (*n* = 176 patients, *p* = 0.064 (FUP period up to eight years) (Figure 3A) [23].

### 3.2. CTC Detection

At baseline, CTCs (≥ 1CTC/7.5 mL blood) were detected in 25.5% of all patients (14/55). 25.0% (9/36) of curative (mean 2.98; median 2.00; range 1–6) and 26.3% (5/19) of palliative PDAC patients (mean 3.60; median 2.00; range 1–11) showed CTCs in their peripheral blood (Figure 2). There was no significant difference in detectability between curative and palliative PDAC patients at baseline. Baseline CTC detection showed no significant association with clinicopathological parameters in curative or palliative patients (Appendix A). Curative patients had (UICC stages I-III) that were surgically resectable, whereas palliative patients presented either M1 (*n* = 16 UICC stage IV) or were not surgically resectable (*n* = 3 UICC stage III patients). Therefore, the UICC classification was used instead of M0/1 for defining the curative and palliative study groups. After a median FUP period of 11 months (range 0–25), an FUP CTC detection rate of 45% (*n* = 10/22) was observed in curative patients in our study cohort. From 14 of 36 curative patients, we could not collect FUP blood samples due to reduced physical status under treatment or perioperative death.

The detection of CTC in baseline blood was not significantly associated with poor prognosis in either the palliative or the curative cohort, although a non-significant impact of baseline blood positivity on shorter RFS for the curative group (11 vs. 9 months; *p* = 0.229) was observed. The most outstanding finding was the significant impact of high CTC counts detected at FUP (during or after chemotherapeutic treatment) on short RFS (≥3 CTCs: *p* = 0.002 (median RFS: 4 vs. 11 months)) (Table 1 and Figure 3B) for the curative cohort, indicating that the presence of CTC in peripheral blood after surgery is more closely associated with an early relapse than base line blood analyses.

### 3.3. Clinical Value of RARRES1 Detection on CTCs

In the palliative group of patients, 75% of all patients’ CTCs were RARRES1 positive; in the curative patient group, 63.6% of patients’ CTCs were RARRES1 positive (*p* = 0.839). RARRES1 protein expression was stable and homogenous on all CTCs of a patient during the disease course. No change of expression was observed under treatment within our patient collective.

The combined analysis of the RARRES1 expression on CTCs and the CTC FUP analysis showed a small potential effect regarding more robust prognostic effect among curative PDAC patients than the CTC analysis alone: Patients with RARRES1 positive CTCs (*n* ≥ 3) detected during FUP had a significantly shorter RFS of 4 vs. 11 months (*p* = 0.001) (Table 1 and Figure 3C).

## 4. Discussion

PDACs are usually detected late and often respond very poorly to the standard chemotherapy regimens given to PDAC patients [3,4,5,18]. Thus, in this study, we first screened for additional markers for better detectability and prognostic power in CTC analyses in PDAC patients. For this, a PDAC cell line and its chemoresistant counterpart (same genetic background) were used as surrogate markers for aggressiveness [19]. Our proteomic survey coupled with mass spectrometry identified RARRES1 as a potential novel marker for PDAC patients—especially in patients with aggressive PDAC after chemotherapeutic treatment. Additionally, we established a multicolor FISH analysis to detect CTC in PDAC patients’ blood using a marker-independent microfluidic CTC enrichment system.

Our data underline that CTC detection at baseline remains challenging and that further improvement of technological approaches is required to increase the detection rates of 25% and 26.3% from the curative and palliative PDAC patient groups to translate this liquid biopsy approach into the clinic for routine use in patient care. Nevertheless, our detection rates align with those reported in previous studies on CTCs in PDAC [6,12,13,14]. The addition of RARRES1 did not increase the detectability of CTCs. In contrast to the baseline results, CTC detection during FUP was more promising (~45%) and might be more suitable for clinical use.

In our patient cohort—neither in the curative nor in the palliative group -baseline CTC detection had a significant impact on OS. This might be caused by small case numbers and a relatively short FUP period.

A significant impact of CTC detection on shorter RFS was observed for the FUP blood analyses in resected patients. The latter indicates that the CTC liquid biopsy approach might be especially suitable for curative patients during FUP to indicate short RFS after resection since detectability and prognostic impact were higher than baseline.

Data from other tumor entities have shown that liquid biopsy could be helpful to stratify patients and adjust the therapeutic options according to the molecular cancer characteristics [24]. E.g., in breast cancer, detection of HER2 positive CTCs in FUP blood of patients with initial HER2 negative tumors is associated with poor OS. Therefore, those patients might benefit from an anti-HER2 targeted treatment [25]. Functional testing of CTCs in PDAC, such as detecting and quantifying metastasis-associated traits, is in its early days [18]. Still, patient management might benefit from such approaches in future settings. Our data showed that the combined analysis of CTC count detection during FUP and an abundant RARRES1 expression on CTCs showed a potential effect of being an even more unfavorable prognostic marker than the CTC detection alone. RARRES1 expression might indicate chemoresistance and aggressiveness in PDAC—and therefore be a marker for poor RFS. More extensive studies, including more patients with different treatment protocols, must be performed to underline this further. Also, functional studies showing the potential mechanistic role of RARRES1 in chemoresistance in PDAC needs to be performed.

RARRES1 is a retinoid acid receptor-responsive gene located at 3q25.32, encoding a type 1 membrane protein and being upregulated by tazarotene and retinoic acid receptors. Although RARRES1 is suggested to inhibit the carboxypeptidase and to be associated with fatty acid metabolism, stem cell differentiation, immunomodulation, and tumorigenesis by regulating the proliferation and migration of tumor cells, the exact mechanisms of action remain unknown and are currently subject to further research studies [26,27,28]. While a hypermethylation-associated downregulation of RARRES1 was observed in several cancers, such as prostate cancer, nasopharyngeal carcinoma, melanoma, and colorectal adenocarcinoma, pointing to a role as a tumor suppressor, RARRES1 expression has been considered an unfavorable prognostic factor in renal cancer and glioma being associated with mesenchymal subtypes [23,26,28,29,30,31]. Nevertheless, to our knowledge, no study has yet investigated the role of RARRES1 in PDAC.

Limitations of the presented study are the small sample size of patients, short FUP period, and reduced number of patients with available FUP samples, which causes a lack of statistical power. Also, the study population was not the optimal to study the role of RARRES1 and so few CTCs were identified. To overcome these limitations, further multi-center studies and an improved patient selection would be necessary. Furthermore, improvements in the technology of CTC detection in PDAC are relevant for future use in patient care with higher CTC detection rates. Our previous studies have shown that using the Parsortix™ cell separation system, more CTCs were detectable in NSCLC, which is another challenging solid tumor with very high heterogeneity among CTCs [21,32]. Unfortunately, the latter was not the case for PDAC; thus, further studies to improve technologies for CTC detection are required to develop a suitable CTC liquid biopsy approach for clinical use in PDAC patients.

## 5. Conclusions

Despite limitations of CTC detection in PDAC, our study highlights the potential of detecting high CTC counts during FUP in resected patients as an unfavorable prognostic marker. Furthermore, for the first time, RARRES1 was analyzed in combination with CTC detection, indicating a possible benefit of adding it as a novel negative prognostic marker for detection of patients with a more aggressive disease. Further knowledge of molecular mechanisms of RARRES1 may presumably support the identification of new therapeutic targets for PDAC and other cancers.

## Figures and Tables

**Figure 1 cancers-14-04405-f001:**
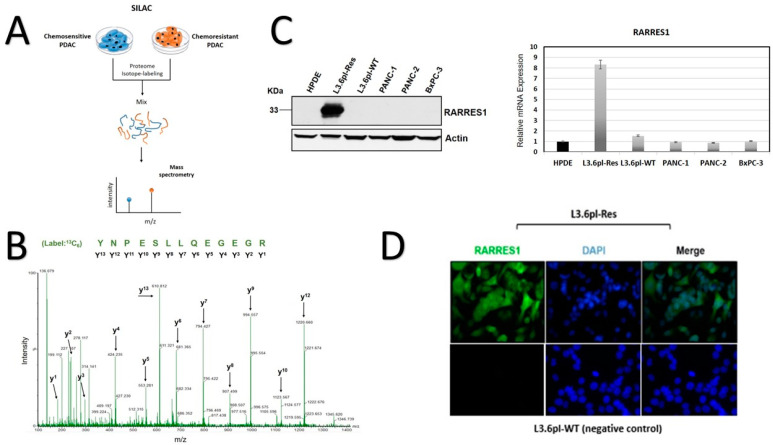
RARRES identification in PDAC cells. (**A**) Schematic representation of SILAC workflow. (**B**) Representative spectrogram from MS/MS analysis that was identified as RARRES1. Peptide-sequence is shown in green. (**C**) Western blot analyses showing no RARRES1 protein expression in normal pancreas cells (HPDE) and different untreated PDAC cell lines. RARRESS1 was only detected in the L3.6pl-Res cells. Actin was used as loading control (left). Real-time RT-PCR results of RARRES1 mRNA expression of the same cell lines (right). The uncropped blots are shown in Appendix A. (**D**) Immunocytochemistry of RARRES1 subcellular localization in L3.6pl-Res cells. DAPI was used for co-staining nuclei.

**Figure 2 cancers-14-04405-f002:**
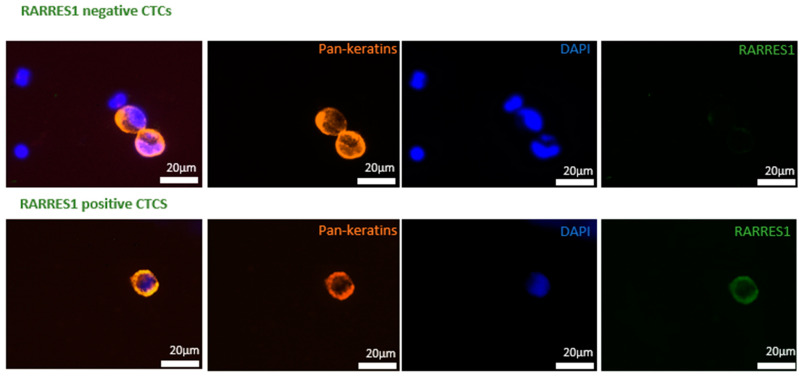
RARRES negative and positive CTCs, with cytokeratine, DAPI, and RARRES1 staining. CTCs detected via immunofluorescence staining. The upper panel shows tow CTCs with no protein expression of RARRES1, whereas the lower panel shows a single RARRES1 positive CTC.

**Figure 3 cancers-14-04405-f003:**
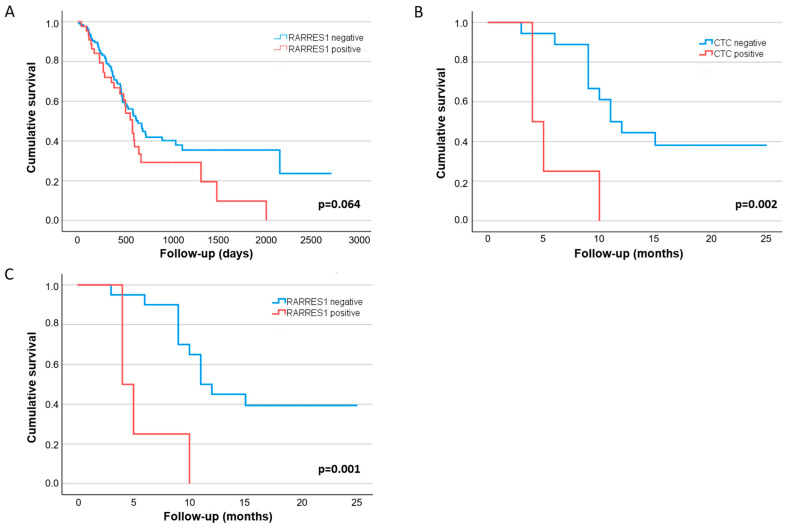
Kaplan-Meier curves of overall (OS) and recurrence-free (RFS) survival in primary tumor tissue and on CTC in PDAC patients. (**A**) OS based on RARRES1 protein expression data of tumor tissue available from The Humsn Protein Atlas (v21.1.proteinatlas.org); https://www.proteinatlas.org/ENSG00000118849-RARRES1/pathology/pancreatic+cancer#ihc (accessed on 15 June 2022); (**B**) RFS based on CTC positive versus negative—stratified by CTC counts (≥3CTCs/7.5 mL EDTA blood) analyzed after the surgery during FUP in curative patients from our study cohort; (**C**) RFS based on RARRES positive versus negative—stratified by the combined positive RARRES status and CTC status (≥3CTCs) during FUP in curative patients from our study cohort.

**Table 1 cancers-14-04405-t001:** Univariate analysis of recurrence-free survival in curative patients.

Univariate		N = 33 ^$^	Median RFS, Months (95% CI)	*p*-Value
Analyses				
Age	≤67 years	17	10.0 (7.2–12.8)	0.218
	>67 years	16	15.7 (11.0–20.5) *	
Gender	male	16	9.0 (3.8–14.2)	0.170
	female	17	15.0 (9.5–20.5)	
ECOG	0	20	-	0.296
	1	12	-	
	2	1	-	
UICC stage	I-II	26	16.1 (12.6–19.6) *	0.005
	III	7	6.0 (3.4–8.6)	
R-status	R0; CRM-	17	10.0 (6.0–14.0)	0.227
	R0; CRM + /R1	16	16.0 (9.1–22.9)	
Grading ^~^	G2	21	11.0 (9.6–12.4)	0.353
	G3	9	14.8 (10.3–19.4) *	
Neoadjuvant treatment	no	26	12.0 (6.0–18.0)	0.089
	yes	7	9.0 (0–19.3)	
Adjuvant treatment	no	4	6.0 (0.0–12.0) *	0.009
	yes	29	12.0 (6.7–17.3)	
Clavien-Dindo	0–2	20	16.0 (7.9–24.1)	0.161
	3–4	13	10.0 (7.7–12.3)	
Ca 19-9	<500 U/ml	24	12.0 (5.9–18.1)	0.881
	≥500 U/ml	9	11.0 (8.2–13.8)	
CTC count per 7.5 mL detected at FUP ^#^	<3	18	11.0 (8.2–13.8)	0.002
	≥3	4	5.8 (2.9–8.6) *	
Combined analysis RARRES1 status and CTC count per 7.5 mL detected at FUP ^#^	<3 OR RARRES1 negative	18	11.0 (8.8–13.2)	0.001
	≥3 AND RARRES1 positive	4	5.8 (2.9–8.6) *	

Univariate analyses (*p*-values, log-rank test). * median not reached; mean was used; **^#^** FUP number of patients ^$^ *n* = 3 perioperatively deceased patients not included in the analyses; ^~^ grading not available for *n* = 3 patients OS, overall survival; CI, confidence interval; ECOG, Eastern Cooperative Oncology Group; UICC, Union for International Cancer Control; CRM, circumferential resection margin; Ca 19-9, Carbohydrate Antigen 19-9.

## Data Availability

The data can be shared up on request.

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
