# Peer review of "Characterization of RARRES1 Expression on Circulating Tumor Cells as Unfavorable Prognostic Marker in Resected Pancreatic Ductal Adenocarcinoma Patients"

_cancers, 2022, doi:10.3390/cancers14184405_

Round 1

Reviewer 1 Report

Comments for Cancers-1830434

 In the manuscript, “RARRES1 expression on circulating tumor cells is an unfavorable prognostic marker in pancreatic ductal adenocarcinoma patients”, Nitschke and co-workers proposed that retinoic acid receptor responder 1 is a novel marker on circulating tumor cells in chemoresistant pancreatic ductal adenocarcinoma cells and early relapse. This scientific hypothesis is reasonable. However, the authors failed to identify this hypothesis. This paper is not ready to publish yet.

Also, some suggestions are listed below for the authors’ consideration.

1.       To establish logic from hypothesis-results-title;

2.       To design more experiments;

3.       To change the writing format, for example, Figure 2 placed to MATERIALS & METHODS is inappropriate for a research paper.

Reviewer 2 Report

In this manuscript, the authors did a novel and interesting study, identifying RARRES1 as a potential aggressive/resistant marker of pancreatic cancers or pancreatic CTCs, which can be potential used as CTC identification markers. however, there are some points to be addressed to improve paper quality.

major:

1. in figure 1D, authors should provide some images for negative control cell lines.

2. Table 1 is too big while contributes little information for the whole manuscript, I would suggest to move to supplemental tables

3. in table 2, authors did a comprehensive univariate analysis with all the factors. As a main finding of RARRES1 and CTC, I suggest authors to generate another seperate table to specify how many CTCs and PARRES1 positive CTCs, PARRES1 negative CTCs are identified in each patient for all 22 patients, are there any correlations between CTC features with any of clinical parameters? UICC stage, treatment, recurrence free survival?

minor:

1. in method part, ‘:” should be removed in line151,160,165, 187

2. authors should provide more detailed information on antibodies (concentration, resource, company, etc) in ‘western blot’ part and ‘immunocytochemistry’ part

3. figure 2 should be in result section instead of method section.

4. overall authors should recheck carefully with punctuations and grammar, structures, etc

overall I would suggest major revision.

Reviewer 3 Report

C. Nitschke, B.Markmann, M. Tölle  et al. study used a microfluidic-based approach for circulating tumor cell (CTC) detection in 55 pancreatic ductal adenocarcinoma (PDAC) patients before treatment initiation (baseline) and during follow-up (FUP). The authors assessed the expression of retinoic acid receptor responder 1 (RARRES1) on CTCs. CTCs were detected in 25.5 % of patients at baseline, while the detection rate during FUP was higher (45.5%). Especially high CTC counts during FUP in resected patients were associated with early tumor relapse. Combining CTC detection and RARRES1 protein expression showed that RARRES1-positive patients with high CTCs counts during FUP had a worse prognosis.

Some points were not addressed.

In the MATERIALS & METHODS section, authors must indicate the sequence of primers that were used in Real-time RT-PCR

It is necessary to indicate the clinical and pathological characteristics of the patients included in the study. The authors did not indicate the stage of PDAC patients.There are no data on the number of patients with stage IV.

In Table 1 (Correlation of CTC detection at baseline with clinicopathological parameters in curative and palliative patients), the authors indicate groups T1-T2; T3-T4; N0; N+, while there is absolutely no data for groups M0 and M1.

The authors indicate that PDAC patients were enrolled for the study from 10/2019-11/2020. At the same time, the overall survival curve shows a range of 2500-3000 days (6-8 years), which does not correspond to the time period.

The authors write that a high number of CTCs is associated with early tumor recurrence, while it is not clear which level of CTCs is high.

Round 2

Reviewer 2 Report

The manuscript can be accepted with appropriate proofreading.